# First In Vitro Characterization of Salinomycinic Acid-Containing Two-Line Ferrihydrite Composites with Pronounced Antitumor Activity as MRI Contrast Agents

**DOI:** 10.3390/ijms26178405

**Published:** 2025-08-29

**Authors:** Irena Pashkunova-Martic, Joachim Friske, Daniela Paneva, Zara Cherkezova-Zheleva, Michaela Hejl, Michael Jakupec, Simone Braeuer, Peter Dorkov, Bernhard K. Keppler, Thomas H. Helbich, Juliana Ivanova

**Affiliations:** 1Preclinical Imaging Laboratory, Department of Biomedical Imaging and Image-Guided Therapy, Division of Molecular and Structural Preclinical Imaging, Medical University of Vienna & General Hospital of Vienna, Waehringer Guertel 18-20, 1090 Vienna, Austria; joachim.friske@meduniwien.ac.at (J.F.);; 2Institute of Inorganic Chemistry, Faculty of Chemistry, University of Vienna, Waehringer Strasse 42, 1090 Vienna, Austriamichael.jakupec@univie.ac.at (M.J.); bernhard.keppler@univie.ac.at (B.K.K.); 3Institute of Catalysis, Bulgarian Academy of Sciences, Acad. Georgi Bonchev Str., Bldg. 11, 1113 Sofia, Bulgaria; daniela@ic.bas.bg (D.P.); zzhel@ic.bas.bg (Z.C.-Z.); 4Institute of Analytical Chemistry, Department of Natural Sciences and Sustainable Resources, BOKU University, Muthgasse 18, 1190 Vienna, Austria; simone.braeuer@boku.ac.at; 5Research and Development Department, Biovet Ltd., 4550 Peshtera, Bulgaria; p_dorkov@abv.bg; 6Faculty of Medicine, Sofia University “St. Kliment Ohridski”, Kozjak Str., 1, 1407 Sofia, Bulgaria

**Keywords:** salinomycin, ferrihydrite, Mössbauer spectroscopy, theranostic agent, high-field MRI

## Abstract

Iron(III) (Fe(III)) complexes have recently emerged as safer alternatives to magnetic resonance imaging (MRI) contrast agents (CAs), reigniting interest in biomedical research. Although gadolinium Gd(III)-based contrast agents (CAs) have been widely used in MRI over the past four decades, their use in the current clinical routine is severely constrained due to concerns about high toxicity and environmental impact. Research is now focusing on synthesizing safer contrast agents with alternative paramagnetic ions like Fe(III) or Mn(II). MRI CAs with integrated potent therapeutic moieties may offer synergistic advantages over traditional contrast agents in clinical use. The study explored the use of salinomycin-ferrihydrite composites as possible effective ensembles of imaging and therapeutic units in the same molecule, evaluating their anticancer activity and influence on the signal in MRI. The composites were characterized using Mössbauer spectroscopy and ICP-MS for iron content determination. The in vitro relaxivity measurements in a high-field MR scanner demonstrated the potency of the composites as T_2_ enhancers. The antitumor activity of one selected Sal-ferrihydrite composite was tested in three human cancer cell lines: A549 (non-small cell lung cancer); SW480 (colon cancer); and CH1/PA1 (ovarian teratocarcinoma) by the MTT cell viability assay. The new Sal-ferrihydrite composite showed a pronounced cytotoxicity in all three human cancers in line with enhanced signal in MRI, which makes it a promising candidate for future biomedical applications. The superior cytotoxic effect, together with the strong signal enhancement, makes these compounds promising candidates for further detailed investigations as future theranostic agents.

## 1. Introduction

The recent development of iron (III) (Fe(III)) complexes as safer alternatives to magnetic resonance imaging (MRI) contrast agents (CAs) has sparked renewed interest in this field of research [1,2,3,4]. While gadolinium (Gd(III))-based CAs have been extensively used in MRI during the last four decades, serious toxicity and environmental concerns considerably limit their use in the current clinical routine [5,6]. Given its half-filled d-subshell and long electronic relaxation times, Fe(III) represents an ideal candidate for the design of both longitudinal relaxation time (T_1_) and transverse relaxation time (T_2_) enhancers for MRI [7,8,9,10,11,12,13,14,15,16]. As an essential element, iron is a vital constituent of hemoglobin, myoglobin, and various cytochromes, ensuring oxygen transport, oxygen storage, and energy generation in the human body [3]. Thus, all biological pathways for its regulation and homeostasis in the living organism already exist, making Fe(III) an even more attractive metal center for the development of new and safer MRI CAs [3].

Several low-molecular-weight Fe(III) complexes have been studied as MRI probes, most of them comprising polyaminocarboxylate ligands or derivatives of them, such as ethylenediaminetetraacetic acid (EDTA) or cyclohexanediaminetetraacetic acid (CDTA) [3]. However, recent reports on the chemistry of this class of Fe(III) coordination compounds showcased several limitations of these ligand scaffolds, resulting in the formation of µ-hydroxide or µ-oxo-bridged species at physiological pH that significantly decrease their relaxivity in MRI [1,2,3]. Kras et al. 2024 [4] reported another type of Fe(III) MRI probe containing a diamino scaffold attached to carboxylate and phenolate donors with no inner-sphere coordinated water with a low longitudinal r_1_ relaxivity. Another challenge in the chemistry of Fe(III)-based MRI CAs that must be considered is the tendency of the Fe(III) center to form six-coordinated complexes having inner-sphere water ligands that exchange rather slowly, leading to reduced MRI relaxivity [7,8].

Previously published data demonstrated that the natural polyether ionophorous antibiotic salinomycin (Sal) (Figure 1) reacted with Fe(III) chloride to form either polynuclear Fe(III) complexes [17] or composites with two-line ferrihydrite [18]. Sal has been shown to selectively eradicate breast cancer stem cells CSCs [19], a finding which was eventually corroborated in other malignancies, such as gastric cancer [20], prostate cancer [21], colon cancer [22], lung adenocarcinoma [23], and leukemia [24]. The ability of Sal to form complexes with metal ions of different valences could be used to further improve its biological activity [25,26,27,28]. It has been demonstrated that one of the composites of Sal with ferrihydrite exerted a pronounced antitumor effect on cervical cancer (HeLa), non-small cell lung cancer (A549), colon cancer (SW480), and ovarian teratocarcinoma (CH1/PA1) cells [18]. Paramagnetic Sal complexes with Gd(III) or Mn(II) show superior antitumor activity against the same A549, SW480, and CH1/PA1 cancer cell lines [28]. The combination of an organic ligand with high antitumor efficiency (such as Sal) with paramagnetic MRI motifs is a modern approach for the synthesis of theranostic agents [28,29,30].

To date, there is no available information about the potential application of complexes or composites of salinomycin with Fe(III) as MRI contrast agents and anticancer drugs in one construct.

This study represents the first in vitro characterization of ferrihydrite-salinomycin composites as MRI contrast agents with significant anticancer activity.

## 2. Results

In our previous study [18], we have demonstrated that the interaction of Fe(III) chloride with salinomycin resulted in formation of composites of two line-ferrihydrite with salinomycin. The composites were characterized by different methods such as elemental analysis, attenuated total reflectance-Fourier transform spectroscopy (ATR-FTIR), electron paramagnetic resonance spectroscopy (EPR), powder x-ray diffraction analysis (XRD), electrospray ionization mass spectrometry (ESI-MS), thermogravimetric analysis with differential thermal analysis (DTA) and mass spectrometry (TG-DTA/MS). The results from the elemental analysis revealed that the composites were of chemical composition [(FeOOH)(C_42_H_70_O_11_)_3_] (composite **1**) and [(FeOOH)(C_42_H_70_O_11_)_3_].4H_2_O (composites **2**, **3** and **4**), respectively. The EPR studies demonstrated that the structural arrangements of the atoms in the composites differed and depended on the synthetic experimental conditions. Composites **1** and **2**, were synthesized at molar ratio between the metal salt and organic ligand 1:1 and contained isolated Fe(III) ions in distorted symmetry and exchanged-bound Fe(III), while the composites **3** and **4** (obtained at molar ratio between the reactants 1:3) consisted mainly of exchanged-coupled Fe(III) ions [18].

Herein, we investigated for the first time the potential of composites **2**, **3** and **4** [18]; as MRI contrast agents with antitumor activity. Since composite **1** was synthesized with the lowest yield [18], it was not included in the present research. In this study, we also carried out Mössbauer spectroscopy for detailed characterization of the iron ions present in the structures of the composites.

### 2.1. ICP-MS Analysis

The determined Fe concentrations by ICP-MS were 0.016, 0.013, and 0.023 mg/L for composites **2**, **3**, and **4** (Table 1), respectively, which is approximately 2% of the total masses of the complexes. There was a good agreement between the experimental values and the theoretical values.

### 2.2. Dynamic Light Scattering Analysis (DLS)

The hydrodynamic diameters of composites **2** and **3** in aqueous solutions were 338.9 nm and 420.1 nm, respectively. For composite **4**, two peaks with intensities of 94.8 and 5.2% were observed, corresponding to hydrodynamic diameters of 369.8 and 102.1 nm, respectively (Table 2). The split signal of the triplicate measurements for composite **4** indicates the presence of two colloidal species with different sizes at the respective concentrations. The average ζ-potential was measured at 19.2, 22.0, and 20.0 mV in triplicate for composites **2**, **3** and **4** (Table 2).

### 2.3. Mössbauer Spectroscopy

Mössbauer analysis determines the parameters of each iron ion in strong dependence on its environment. The quadrupole splitting (QS) values for Db1 were 0.49, 0.62, and 0.54 for composites **2**, **3**, and **4**, respectively, whereas the measured values for Db2 varied from 0.87 mm/s (composite **2**) to 1.06 mm/s (composite **3**) (Figure 2; Table 3). The estimated isomer shifts (IS) for all three compounds were similar and resulted in 0.37 to 0.43 mm/s for Db1 and Db2 of composites **2**, **3**, and **4** (Table 3).

### 2.4. In Vitro MRI Relaxivity

The magnetic susceptibility of all three composites of Sal was determined in a dilution series in MeOH and compared to pure methanol as a negative control. In addition, all three compounds were first dissolved in pure DMSO and subsequently diluted with Milli-Q^®^ water to assess their effect on the proton relaxation in aqueous medium. Milli-Q^®^ water was used as a negative control. For composite **2**, a steep signal increase with growing concentrations was observed for the relaxation rate R_1_, as well as for the R_2_ values of composites **2** and **3**. (Figure 3). Composite **2** showed an increased longitudinal r_1_ relaxivity, with 0.94 1/mM·s in MeOH and 2.41 1/mM·s in H_2_O compared to composites **3** and **4**, with values for r_1_ of 0.06 and 0.09 1/mM·sec measured in methanol, respectively, and 0.30 and 0.28 1/mM·s in H_2_O (Table 4, Figure 3). A high transversal relaxivity r_2_ for composite **2** was also observed, which remained unchanged in aqueous medium (r_2_ = 8.41 and 8.44 1/mM·s in methanol and water, respectively). Composites **3** and **4** exerted superior r_2_/r_1_ ratios with r_2_ 8.68 and 2.01 1/mM·s in methanol. However, a significant drop of r_2_ for composite **3** in water was shown, whereas composite **4** exerted an approximately four-fold increase in its transversal relaxivity in aqueous medium, with r_2_ = 7.63 1/mM·s (Table 4, Figure 3).

The proton relaxivities of all three Sal-composites are summarized in Table 4. Literature data for other Fe(III)-based contrast agents are also given for comparison.

### 2.5. Cytotoxicity Studies

Due to the incomplete dissolution of composites **2** and **3** in DMSO, only composite **4** was further evaluated in cell culture studies. Its cytotoxic activity was tested in three human cancer cell lines. The composite exerted enhanced antitumor activity in all cell lines compared to the free salinomycinic acid (SalH) and SalNa (Figure 4, Table 5). The increase in cytotoxicity was roughly two- to six-fold, depending on the cell line, yielding IC_50_ values between 0.071 µM (CH1/PA-1 cells) and 0.53 µM (SW480 cells). The effect of the composite on cell viability was more pronounced on CH1/PA-1 and A549 cells, corresponding to the profile of SalH and SalNa, but, as the increase was stronger in CH1/PA-1 cells, this broadly chemo-sensitive cell line was shown to be the most sensitive (Figure 4, Table 5).

## 3. Discussion

A recent study revealed that ferrihydrite-containing nanoparticles exerted significant negative (T_2_-weighted) contrast and could be used as a good model of natural ferritin in studies of its MRI properties [29]. The synthesized nanoparticles demonstrated low magnetism and good contrast accompanied by good biocompatibility and low toxicity. To the best of our knowledge, there is a gap in the literature about the potency of composites of ferrihydrite with organic compounds with antitumor activity as contrast agents for MRI. Such composites are of interest, since they could have potential as theranostic probes.

Herein, we prepared and investigated two-line ferrihydrite composites of Sal as contrast agents and anticancer drugs in one. The Mössbauer analysis, carried out in this study, determines the parameters of each iron ion in strong dependence on its environment. The high sensitivity of the method on one hand and the inhomogeneity and geometric differences on the other lead to the superposition of multiple spectra, which overlap and give a complex profile with a broadening of the Lorentzian line of spectra. Based on previous studies of the composites [18], two types of Fe(III) ions have been identified—isolated and exchange-bound. In the Mössbauer spectra, they are registered in a common region due to the similarity in the parameters of the high-spin state and octahedral coordination of both types. The isomer shifts (Table 3) were close to the isomer shift value for ferrihydrite reported by Murad [30] and corresponded to octahedral high-spin Fe(III) ions. The isomer shift in Db 2 in the spectrum of composite **2** is higher than the other composites and generally than the values cited in the literature for iron hydroxides [33]. The quadrupole splitting (QS) values for Db2 varied from 0.87 mm/s (composite **2**) to 1.06 mm/s (composite **3**). It has been reported that, for paramagnetic ferrihydrites, the QS is higher compared to QS for other Fe(III) oxides and is due to Fe(III) in distorted symmetry [31]. The QS for Db1 revealed the presence also of Fe(III) with a more symmetric charge distribution. The determined values for the quadrupole splitting were highest for composite **3** and lowest for composite **2**. Most likely, the observed lower values for QS for composite **2** compared to the values for QS for composites **3** and **4** are related to the presence of isolated Fe(III) ions and magnetically coupled Fe(III) [18].

Particle size, or hydrodynamic diameter, is a major determinant for further biological studies and future in vivo applications. Monodisperse particles with sizes between 338 and 420 nm and a low degree of agglomeration suggested by the zeta potential analysis, as demonstrated with the 2-line ferrihydrite composites of Sal, might be optimal for blood-pool imaging agents in MRI [34]. The surface charge of the colloidal solutions of our composites, measured as a ζ potential, is sufficient to repel neighboring particles, thus holding the suspension in the form of individual particles, and lies between 19.2 and 22.0 mV. In this case, a suspension of charged individual particles retains nanoscale characteristics, such as a large surface area. Particles with minimal charge, on the other hand, may clump together to form aggregates, which have the same total mass as the particle suspension but a significantly smaller surface area.

The r_1_ relaxivity values of the composites of salinomycin with two-line ferrihydrite were similar to those found for reported low-molecular iron-based coordination compounds [3,7], whereas the estimated r_2_ values of composites **2** and **4** within this study were the most superior published thus far (Table 4). Chen et al. summarized mono- and multinuclear Fe(III) complexes, comprising open ligands such as N,N′-bis(2-hydroxyphenyl)ethylenediamine-N,N′-diacetic acid (HBED) and derivatives, or trans-cyclohexane diamine tetraacetic acid (tCDTA) [3]. The relaxivities r_1_ and r_2_ for the latter, for instance, were 2.2 and 2.5 mM·s^−1^ at 0.94 T, which were measured in serum. Macrocyclic Fe(III) complexes with alcohol donor groups were reported previously by Snyder et al. and showed a good T_1_ effect in buffer, though lacking a coordinated inner-sphere water [7]. It should be noted that two relaxivity measurements in water were performed in this study for each composite using the same sequence as described in Section 4, but with different sequence parameters. For R_1_, the parameters considered the partially low T_1_ values, whereas, for R_2_, a longer TR was applied not only because of the influence of T_1_ but also to allow T_2_ times longer than 80 ms. The MR behavior of contrast agent compounds depends on the ratio of the relaxivities (r_2_/r_1_). Should the material act as a T_2_ CA, r_2_/r_1_ is greater than 10, and if it acts as a T_1_ contrast agent, r_2_/r_1_ < 2 [14]. Thus, composites **2** and **3** with r_2_/r_1_ ratios of 3.50 and 6.07 may serve as dual T_1_/T_2_ signal enhancers, whereas composite **4** may be efficient only as a T_2_ or negative contrast agent in MRI. Although the composites have an identical chemical composition, their structures differed significantly. The results from the analysis with electron paramagnetic spectroscopy (EPR) [18] proved that composite **2** contained isolated Fe(III) and exchanged-coupled Fe(III). Most likely, the isolated Fe(III) in a high-spin state contributed to the observed elevated r_1_ value for composite **2** both in methanolic and aqueous media compared to composites **3** and **4**. When the proton environment was changed, as in the dilution series with water, an expected relaxivity increase was observed for all composites, ranging from 2.6- to 5-fold for composites **2**, **4**, and **3**, respectively (Table 4). It was interesting to observe that composite **4** exhibited an approximately four times lower T_2_ effect in methanol compared to composite **3** and just the opposite when the measurements were carried out in water. It should be noted that different Mössbauer parameters for all three composites were also established. The difference in the Mössbauer characteristics and r_2_ relaxivities between composite **3** and composite **4** most likely was related to the different proportions of isolated and exchange-coupled Fe^3+^ ions, as well as interaction strength. Although composite **4** exerted a much weaker effect on both T_1_ and T_2_ in methanol compared to the other two composites, it showed a 2.4-fold increase of r_1_ and a strong effect on the T_2_ relaxation time in water with r_2_ = 7.63 mM·s^−1^. The r_2_/r_1_ ratio of 27.3 establishes it as a potentially efficient T_2_ enhancer with r_2_ transverse relaxivity superior to the reported low-molecular-weight iron complexes [3,4,5,6,7]. With regard to iron oxide-based nanoparticles [33], the observed r_2_ relaxivity is comparable or lower to those, depending on the medium and the applied magnetic field strength. Compared to the latter [33], the composites demonstrated either measurable or lower T_2_ effects (Table 4). It should be noted, however, that iron nanoparticles have low liver clearance [7]. Lu et al. demonstrated a very high relaxivity r_1_ of 6.31 mM·s^−1^ achieved with ultrasmall iron oxide nanoparticles applied as a contrast agent for magnetic resonance angiography [34]. We demonstrated a high r_1_ relaxivity only with composite **2** measured in an aqueous medium with r_1_ = 2.4 mM*s^−1^, which is comparable to the clinically applied Gadolinium-based Magnevist^®^ (Bayer HealthCare Pharmaceuticals Inc., Leverkusen, Germany) with r_1_ = 3.3 at 3T magnetic strength field [35].

Compared to Fe(III) complexes with low molecular weight ligands [3,4,5,6,7], presented in Table 4, composites **2** and **4** in Milli-Q^®^ water demonstrated a stronger T_2_ effect. Published data revealed that transverse relaxivity of native ferritin varied from 1 to 10 mM·s^−1^ and was dependent on the magnetic field strength [35]. In our study, we observed high r_2_ values at 9.4 T magnetic field strength. In the new composites of salinomycin with two-line ferrihydrite, most likely magnetically coupled Fe(III)–Fe(III) interactions predominate, which results in an enhanced T_2_ effect. Moreover, the attachment of three macromolecular Sal-ligands may contribute to a slow molecular tumbling and to a decreased rotational motion τ_R_ of the entire assembly, which could further enhance relaxivity values.

Based on the results presented in this study and literature data [1,9,31,36] it could be summarized that the high-spin Fe(III) complexes, containing isolated Fe(III) ions only are potential T_1_ contrast agents, while compounds consisted of Fe(III) oxo(hydroxides) with exchanged coupled Fe(III) ions exert increased T_2_ effect. The combination of a biocompatible form of iron (ferrihydrite) in our study, with a ligand with broad antitumor activity at the submicromolar range, could provide many potential theranostic applications.

It was interesting to observe that the antitumor activity of composite **4** was much higher compared to the antitumor activity of the first composite of salinomycinic acid and ferrihydrite reported by Ivanova et al. 2024 [18]. These results can be explained by the different structures of both composites. Composite **4** contained mainly exchange-coupled Fe(III) [8]. Most likely, this binding mode of Fe(III) contributed to the enhanced r_2_ relaxivity and increased cytotoxic activity of this composite. Composite **4** exerted about two times higher antitumor activity in A549 and CH1/PA-1 cells compared to Mn(II) salinomycinate (Table 5). Even the cytotoxic activity of Gd(III) salinomycinate on both cell lines was lower compared to that of the two-line ferrihydrite composite with salinomycin [18]. Moreover, composite **4** demonstrated a superior anticancer activity compared to cisplatin. In addition, the prepared Sal-ferrihydrite exhibited a strong cytotoxic effect in colon carcinoma cells (SW480) that was comparable to that of the Sal-Mn complex and more pronounced compared to cisplatin [37] (Table 5).

## 4. Materials and Methods

### 4.1. Chemicals

Salinomycin sodium (C_42_H_69_O_11_Na; SalNa) was a gift from Biovet Ltd. (Peshtera, Bulgaria), purity > 95%. Organic solvents (MeCN, MeOH, DMSO) and FeCl_3_.6H_2_O of analytical grade were supplied by Fisher Scientific (Loughborough, UK). MTT (3-(4,5-Dimethylthiazol-2-yl)-2,5-Diphenyltetrazolium Bromide) was provided by Sigma-Aldrich (Vienna, Austria). Fe and In analytical standards were obtained from LabKings (Hilversum, The Netherlands). Nitric acid (Rotipuran^®^ Supra 69%) was purchased from Carl Roth GmbH (Karlsruhe, Germany).

### 4.2. Synthesis

The experimental procedures for the synthesis of the composites were previously described by Ivanova et al. 2024 [18]. Briefly, composite **2** was synthesized as follows. Two mL metanolic solution of 0.5 mmol Fe(III) chloride hexahydrate solution was added to 5 mL 0.5 mmol solution of salinomycin sodium (SalNa) in methanol. The reaction mixture was stirred for 30 min. The solvent was evaporated at room temperature to a final volume of 2 mL and the solid phase was isolated by filtration, thoroughly washed with deionized water, and dried over P_2_O_5_. For the synthesis of composite **3**, 2 mL aqueous solution of 0.3 mmol Fe(III) chloride hexahydrate was mixed with 0.9 mmol salinomycin sodium solution, prepared in CH_3_CN:CH_3_OH (1:5) (2 mL CH_3_CN and 10 mL CH_3_OH). After stirring the reaction mixture for 30 min, the solvents were evaporated to a final volume of 2 mL at room temperature. The precipitate was isolated, washed with deionized water, and dried in a desiccator for seven days. The synthesis of composite **4** was conducted at a metal salt-to-ligand molar ratio of 1 to 3. The ligand and the metal salt were dissolved in 5 mL and 2 mL CH_3_OH, respectively.

All composites were of the chemical composition [(FeOOH)(C_42_H_70_O_11_)_3_].4H_2_O and had a molecular weight MW = 2413.944 g/mol [18]. Details about the yield, purity, and chemical characterization of the composites were presented in our recent publication by Ivanova et al. 2024 [18].

### 4.3. Determination of the Fe Content by Inductively Coupled Plasma Mass Spectrometry (ICP-MS)

Five milligrams of each composite compound were dissolved first in concentrated nitric acid (69% HNO_3_, Rotipuran^®^ Supra), digested by an automated thermal program, and then subjected to ICP-MS for the determination of the Fe content. An ICP-MS 7800 (Agilent Technologies, Tokyo, Japan) was used, with He as a collision gas. Fe was monitored at *m*/*z* 56 and quantified via external calibration. In was added as the internal standard. All solutions were prepared in 3% nitric acid.

### 4.4. Dynamic Light Scattering (DLS)

DLS measurements of the composites’ dilutions in water prior MRI were performed on a Malvern Zetasizer Nano ZSP at 25 °C at a fixed scattering angle of 173°. The sample solutions (0.2 mg mL^−1^) were prepared in ultra-pure MilliQ^®^ water, sonicated for 5 min and filtered through 0.45 µm nylon syringe filters prior to the measurement. The zeta potential (ζ-potential) was determined in the same sample solutions using a Zetasizer Nano-ZS from Malvern (Malvern Panalytical, Malvern, UK) in ultra-pure MilliQ^®^ water as dispersant (viscosity = 0.8872 cP and refractive index, RI = 1.330), calibrated with a Zeta Potential transfer standard (−40 ± 5.8 mV).

### 4.5. Mössbauer Spectroscopy

The ^57^Fe Mössbauer spectra were recorded using the Wissel spectrometer (Starnberg, Germany) in a constant acceleration mode.

The measurements were performed at room temperature (293 K) with an analyzer with 1024 channels and source 57Co/Rh (15 mCi). A two-doublet model was applied in the fitting of the spectra due to the visually asymmetric nature of the experimental lines. The mathematical fits were made with equal line widths, the value of which were determined using the α-Fe standard. The quadrupole splitting (QS) and isomer shift (IS) with respect to α-Fe were estimated using the NORMOS (based on the least squares approach) computer program with Lorentzian line fitting. The measurements of the line width (FWHM), QS, and IS were assessed with an error of ±0.01 mm/s.

### 4.6. In Vitro Relaxivity Measurements

A dilution series of composites **2**, **3**, and **4** were prepared and evaluated for their effect on the proton relaxivity in MRI. For composites **2** and **3**, the concentration range was 0.060–0.015 mM, and for **4**, it was 0.166–0.043 mM Fe. For dilution series, all tested compounds were first dissolved in pure methanol and consequently diluted to a final sample volume of 0.6 mL. As references, pure methanol served as a negative control. In addition, relaxivity measurements were conducted in water to ensure comparison with already published iron-containing contrast agents. All composites were first dissolved in pure DMSO and consequently diluted with Milli-Q^®^ water with a final sample volume of 0.6 mL and in the same concentration range as mentioned above. As a reference, pure Milli-Q^®^ water served as a negative control. The samples were placed in Eppendorf Safe-lock polypropylene (PP) tubes (0.4 cm diameter) in the center of a plastic box at room temperature.

A high-field MRI scanner by Bruker Biospec (9.4 Tesla) and an 86 mm volume coil were used for all measurements. The T_1_ and T_2_ relaxation times for various concentrations of contrast media were evaluated through a rapid acquisition with relaxation enhancement with a variable repetition time (RAREVTR) sequence, as well as a multislice multiecho spin echo sequence (MSME). The parameters for the RAREVTR sequence included an echo time (TE) of 7 ms and a series of repetition times ranging from 200 to 5500 ms (TI values of 200, 400, 800, 1500, 3000, and 5500 ms). Additional specifications included a matrix size of 256 × 256 and a slice thickness of 2 mm. For the MSME sequence, we implemented 12 echoes with a TE ranging from 6.9 ms to 83 ms and a repetition time (TR) of 2600 ms. The matrix size for this sequence was 192 × 192, with a slice thickness of 1 mm. The relaxation times T_1_ and T_2_ were derived using the ParaVision V3.3 built-in image sequence analysis tool (ISA). For all measurements, the relaxation rates (R_1_, R_2_), which are the inverses of the relaxation times, were calculated for each substance. The relaxivities (r_1_, r_2_, and r_2_*) were calculated using the slope of a linear regression of R_1_ and R_2_/R_2_* as a function of contrast agent concentration.

### 4.7. Cytotoxicity Tests

#### 4.7.1. Cell Culture

To evaluate the antitumor activity of the newly prepared Sal-Fe(III) composites, three adherently growing human cancer cell lines were used: A549 (non-small-cell lung cancer); SW480 (colon carcinoma) cells; kindly provided by the Institute of Cancer Research, Department of Medicine I, Medical University of Vienna, Austria; and CH1/PA-1 (ovarian teratocarcinoma) cells, a gift from Lloyd R. Kelland (CRC Center for Cancer Therapeutics, Institute of Cancer Research, Sutton, UK). All cell lines were grown and processed as previously described [18].

#### 4.7.2. MTT Assay

All prepared Sal-ferrihydrite composites are sparsely soluble in water, whereas they have good solubility in methanol, DMSO, and in ethanol. However, after assessing the highest possible amount of the composites to start with, the best solubility in DMSO was achieved for composite **4**. Therefore, only this composite was subjected to biological characterization.

The antiproliferative activity of a compound was determined using the colorimetric MTT assay as published by Ivanova et al. [18]. Briefly, each composite was dissolved in DMSO, diluted in supplemented MEM, and added to 96-well microculture plates, containing 1 × 10^3^ CH1/PA-1, 2 × 10^3^ SW480 and 3 × 10^3^ A549 cells per well, respectively. After a 96 h-exposure, the drug-containing medium was replaced with an RPMI 1640/MTT mixture, supplemented with heat-inactivated fetal bovine serum, and 4 mM L-glutamine. The MTT-containing medium was then replaced with 150 μL DMSO per well to dissolve the formazan product formed by viable cells. The optical density at 550 nm was measured, and the 50% inhibitory concentrations (IC_50_) were interpolated from the concentration-effect curves. Three independent experiments were conducted, with each concentration level tested in triplicate.

## 5. Conclusions

In this study, we demonstrated, for the first time, that the relaxivity values and Mössbauer parameters of composites of salinomycin with two-line ferrihydrite depended on the experimental conditions of their synthesis. Our results proved that the highest transverse r_2_ relaxivity was achieved for composites synthesized in mixed solvents (water for the metal salt), CH_3_OH:CH_3_CN (1:5) for salinomycin with a metal salt:organic ligand ratio of 1:3. The highest r_1_ relaxivity was established for a composite synthesized in a methanolic environment and a metal salt:salinomycin ratio of 1:1. Mössbauer spectroscopic investigation demonstrated advantages of salinomycin composites with two-line ferrihydrite in creating different environments for iron ions, which are in a high-spin state and octahedrally coordinated. The pronounced antitumor activity of composite **4**, together with its significant T_2_ effect, revealed its potential as contrast agent for MRI. Further extensive research is needed to optimize the best nanocarrier for the synthesized composites and to study the pharmacokinetics, biodistribuiton and metabolism of the assemblies in experimental animals.

## Figures and Tables

**Figure 1 ijms-26-08405-f001:**
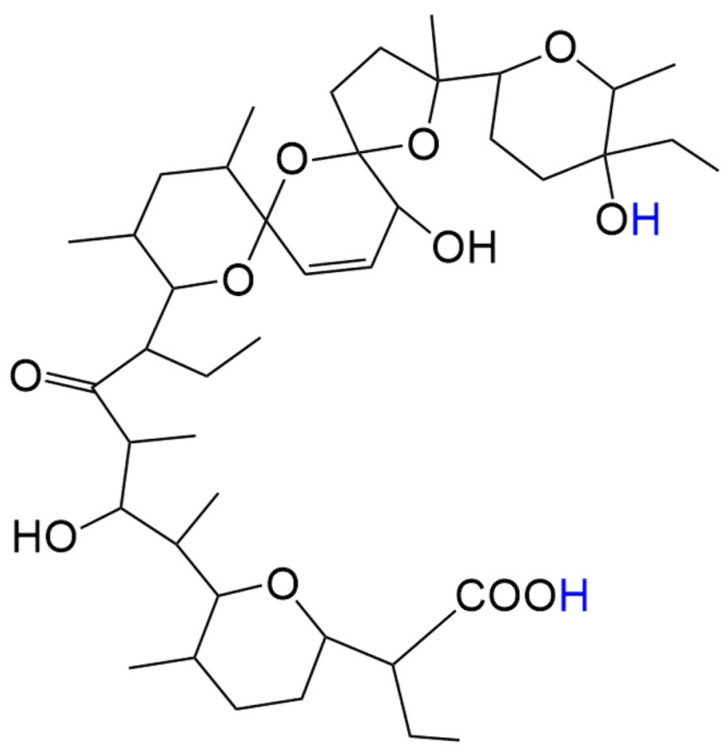
Structure of salinomycinic acid (salinomycin, Sal) with possible hydrogen donor atoms (blue).

**Figure 2 ijms-26-08405-f002:**
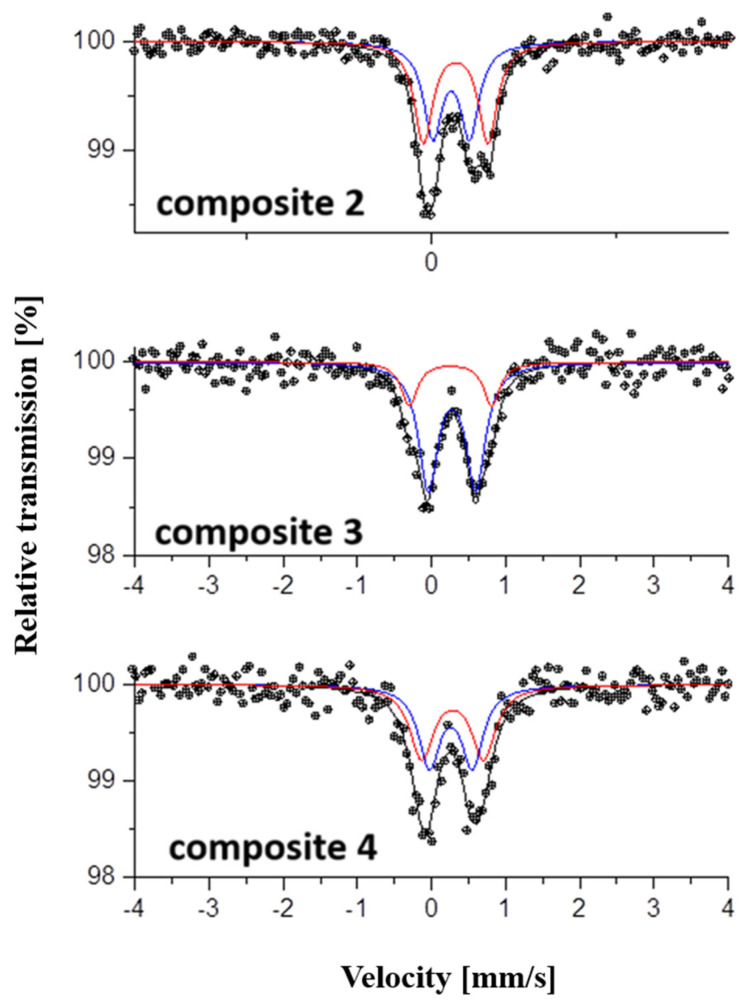
Mössbauer spectra of composites **2**, **3**, and **4**. The blue line corresponds to Db1, the red line depicts Db2, and the black line is the curve-fitting experimental spectrum.

**Figure 3 ijms-26-08405-f003:**
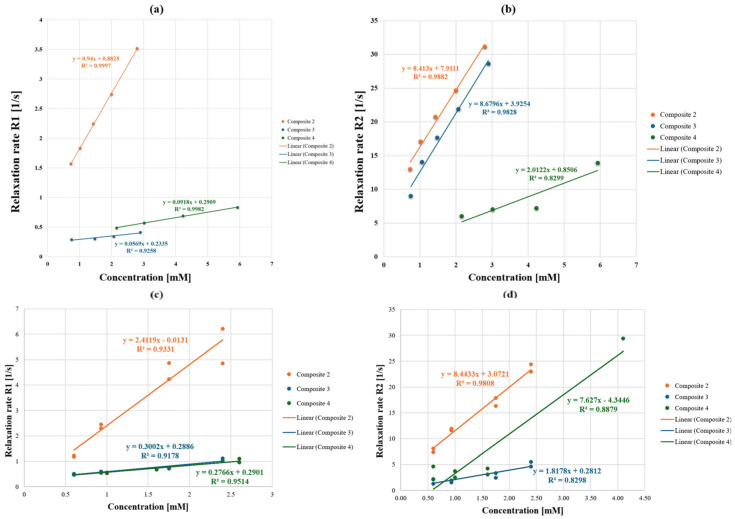
Plots of relaxation rates R_1_, [1/s] (**a**,**c**), and R_2_, [1/s] (**b**,**d**), of two-line ferrihydrite composites with Sal as functions of the concentration [mM]. Dilution series (**a**,**b**) were measured in methanol, and (**c**,**d**) in water. Composite **2** showed a steep T_1_-weighted signal increase (**a**), which was more pronounced in aqueous medium (**c**) and an increased T_2_ signal (**b**). Composites **3** and **4** exerted a dominant T_2_ effect (**b**). The T_2_ effect of composite **2** in water remained unchanged (**c**), whereas that of composite **3** dropped off significantly, while, for composite **4,** a signal increase was shown (**d**).

**Figure 4 ijms-26-08405-f004:**
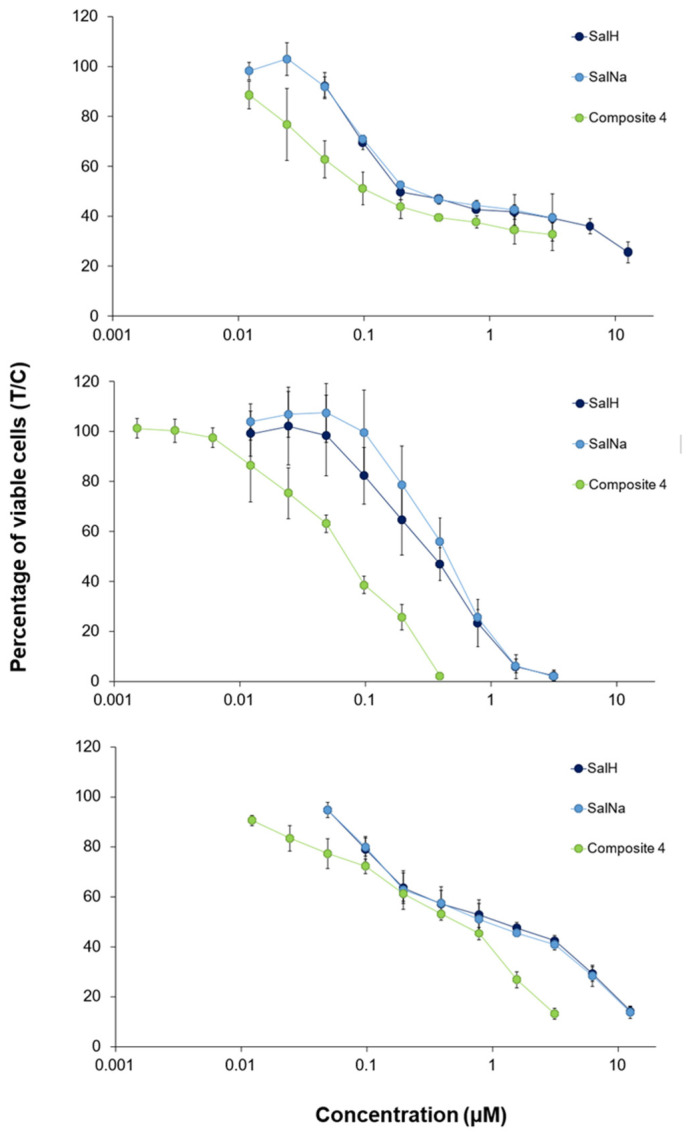
Concentration-effect curves of SalH, SalNa, and composite **4**, assessed via the MTT assay following a 96 h treatment of A549 (**top**), CH1/PA-1 (**middle**), and SW480 (**bottom**) cells. Values, which are normalized in relation to untreated controls, represent means and standard deviations derived from a minimum of three independent experiments.

**Table 1 ijms-26-08405-t001:** Estimated concentrations of Fe in [mg/L] of the prepared two-line ferrihydrite composites **2**, **3**, and **4** by ICP-MS.

Number	Tube	Compound	Fe Concentration, mg/L	Total Fe Concentration, mg/kg	Theoretical Fe Concentration, mg/kg
1	A2	Composite **2**	0.013	24,166	23,200
2	A3	Composite **3**	0.016	21,730	23,200
3	A4	Composite **4**	0.023	20,257	23,200

**Table 2 ijms-26-08405-t002:** DLS data of composites **2**, **3**, and **4**. The composites were diluted as for the MRI experiments first in pure DMSO and consequently with Milli-Q^®^ water (Merck KGaA, Darmstadt, Germany). After an additional 1:10 dilution with Milli-Q^®^ water and sonication, DLS measurement in triplicate was performed: temperature = 25 °C; PCS refractive index (RI = 1.333); viscosity = 0.8872 cP; PDI = 0.4–0.5. For composite **4**, two peaks with intensities of 94.8 and 5.2% were observed, corresponding to hydrodynamic diameters of 369.8 and 102.1 nm, respectively.

Compound	Composition	Aver Size [nm] ± St Dev	Intensity [%]	ζ-Potential [mV] ± St Dev
Composite **2**		338.9 ± 3.94	100	19.2 ± 0.700
Composite **3**	[(FeOOH)(C_42_H_70_O_11_)_3_].4H_2_O	420.1 ± 25.21	100	22.0 ± 0.557
Composite **4**		369.8 ± 34.89	94.8	20.0 ± 0.751
		102.1 ± 1.78	5.2	

**Table 3 ijms-26-08405-t003:** Mössbauer parameters obtained for composites **2**, **3**, and **4**; IS—isomer shift; QS—quadrupole splitting; A—relative area.

Sample	Components	IS, mm/s	QS, mm/s	FWHM, mm/s	A, %
Composite **2**	Db1-Fe(III)	0.37	0.49	0.30	48
	Db2-Fe(III)	0.43	0.87	0.30	52
Composite **3**	Db1-Fe(III)	0.38	0.62	0.30	69
	Db2-Fe(III)	0.37	1.06	0.30	31
Composite **4**	Db1-Fe(III)	0.37	0.54	0.30	57
	Db2-Fe(III)	0.38	0.92	0.30	43

**Table 4 ijms-26-08405-t004:** Comparison of proton relaxivities r_1_ and r_2_ [1/mM·s] of the new composites **2**, **3**, and **4** with other low-molecular-weight Fe(III) complexes and with Fe_x_O_y_-based nanoparticles.

Complex	r_1_ [1/mM·s] MeOH/H_2_O	r_2_ [1/mM·s] MeOH/H_2_O	r_2_/r_1_ MeOH/H_2_O	Magnetic Field Strength, [T]
Composite **2**	0.94/2.41	8.41/8.44	8.95/3.50	9.4
Composite **3**	0.06/0.30	8.68/1.82	144.67/6.07	9.4
Composite **4**	0.09/0.28	2.01/7.63	22.3/27.3	9.4
PAA-Fe@Fe_3_O_4_ [31]	0.72–4.00	13.8–27.8	4.5–38.6	0.5
PEG-Fe@Fe_3_O_4_ [31]	0.72–4.00	13.8–27.8	4.5–38.6	0.5
DHCA-Fe@Fe_3_O_4_ [31]	0.72–4.00	13.8–27.8	4.5–38.6	0.5
Fe(L2) [7]	1.1	1.4	1.16	1.4
Fe(CDTA) [3]	2.0 *	2.2 *	1.1	0.94
Fe(HBED) [3]	0.49	0.52	1.06	1.5

* measured in serum; all cited relaxivity values were determined in aqueous medium.

**Table 5 ijms-26-08405-t005:** Cytotoxicity of composite **4** compared to free salinomycinic acid (SalH) and salinomycin sodium (SalNa), and to the chemotherapeutic drug cisplatin^®^: mean IC_50_ values (in µM) ± standard deviations from at least three independent MTT assays in each of the three human cancer cell lines.

Cell Line	A549	SW480	CH1/PA1
Sample			
1. SalH [28]	0.23 ± 0.06	1.1 ± 0.6	0.32 ± 0.12
2. SalNa [28]	0.27 ± 0.02	0.88 ± 0.44	0.43 ± 0.11
Composite **4**: [(FeOOH)(C_42_H_70_O_11_)_3_].4H_2_O	**0.117** ± **0.062**	**0.53** ± **0.08**	**0.071** ± **0.004**
[Mn(Sal)_2_(H_2_O)_2_] [28]	0.19 ± 0.11	0.52 ± 0.22	0.17 ± 0.05
[Gd(Sal)_3_(H_2_O)_3_] [28]	0.15 ± 0.12	0.36 ± 0.12	0.093 ± 0.025
Cisplatin^®^ [32]	6.2 ± 1.2	3.3 ± 0.2	0.077 ± 0.006

## Data Availability

The original data presented in the study are openly available in Mendeley Data at https://doi.org/10.17632/p4wwk57mhf.2, accessed on 25 August 2025.

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
