# Peer review of "First In Vitro Characterization of Salinomycinic Acid-Containing Two-Line Ferrihydrite Composites with Pronounced Antitumor Activity as MRI Contrast Agents"

_ijms, 2025, doi:10.3390/ijms26178405_

Round 1
Reviewer 1 Report
Comments and Suggestions for Authors
This manuscript reports three Sal-ferrihydrite composites, with iron content and valence state analyzed using ICP-MS and Mössbauer spectroscopy, and longitudinal (r1) and transverse (r2) relaxation rates measured under a 9.4T magnetic field. The composites exhibited significantly different magnetic resonance signal enhancement capabilities under different synthesis conditions. Composite 2 demonstrated higher dual signal enhancement potential for both r1 and r2, while composite 4 exhibited stronger T2 effects and superior antitumor cell activity (IC50 values of 0.071-0.53 µM against A549, SW480, and CH1/PA-1 cells), outperforming free Sal and the clinical drug cisplatin. It was an interesting manuscript. If the authors could fully address the following comments, I'd recommend the publication of the manuscript.
- How did variations in the Fe3+ coordination environment under different synthesis conditions influence the r₁ and r₂ relaxivities? Could ligand design be used to precisely modulate the T₁/T₂ contrast performance?
- Could the use of nanocarriers (such as liposomes) improve the water solubility and bioavailability of the composite, thereby enhancing in vivo imaging performance?
- What were the pharmacokinetics, plasma protein binding profile, and biodistribution of the composite in vivo? Could it achieve tumor-specific accumulation?
- Under high-field (9.4T) and clinically common low-field (1.5-3T) MRI conditions, did r1 and r2 maintain similar trends?
Author Response
Dear reviewer, we would like to thank you for your time to evaluate our manuscript. We appreciate your comments and provide the following responses below, along with the corresponding revisions in track changes in the resubmitted files.
Comments 1: How did variations in the Fe3+ coordination environment under different synthesis conditions influence the r₁ and r₂ relaxivities? Could ligand design be used to precisely modulate the T₁/T₂ contrast performance?
Response 1: Thank you for pointing this out. Although the studied composites had the same chemical composition, the structural arrangements of the atoms differed as demonstrated by EPR studies, reported in our preliminary work (please see reference 18 in the references section of the manuscript). Composite 2 (synthesized at a molar ratio between the metal salt and salinomycin of 1:1) contained both isolated and exchanged−bound Fe(III), while in composites 3 and 4 (both synthesized at a molar ratio between the reactants of 1:3), mainly exchanged−bound Fe(III) was detected. Our data and literature studies revealed that metal complexes containing isolated high-spin Fe(III) could serve as T1 contrast agents. In contrast, compounds containing mainly Fe(III) oxo(hydroxides) with magnetically coupled Fe(III) would be more suitable as T2 contrast materials (please see the discussion section in the revised manuscript, text in track changes).
Concerning ligand design, numerous research studies are being performed with the leading groups of Huczynski et al. and Antoszczak et al., who prepared different salinomycin amide substitutes like n-butylamine, phenylamine, 4′-aminobenzo-15-crown-5, 3-morpholine-propylamine, benzylamine, tryptamine, and 3,6,9-trioxadecylamine, benzotriazole ester of salinomycin (Huczyński, A., Janczak, J., Antoszczak, M., Wietrzyk, J., Maj, E., & Brzezinski, B. (2012). Antiproliferative activity of salinomycin and its derivatives. Bioorganic & medicinal chemistry letters, 22(23), 7146–7150. https://doi.org/10.1016/j.bmcl.2012.09.068; Antoszczak, M., Krzywik, J., Klejborowska, G., Sulik, M., Sobczak, S., Czerwonka, D., Maj, E., Ullrich, M., Sobierajski, T., Sukiennik, J., Wietrzyk, J., Mozga, W., Pilaszek, P., & Huczyński, A. (2025). Effect of stereochemistry at position C20 on the antiproliferative activity and selectivity of N-acylated derivatives of salinomycin. European journal of medicinal chemistry, 291, 117598. https://doi.org/10.1016/j.ejmech.2025.117598). However, the main objective was to improve water solubility by preserving anticancer activity. In all cases, the derivatization of Sal involved the OH group at position C20 and/or the end-standing COOH group. With regard to the synthesis of new MRI contrast agents, the donor H-atoms of the ligand must be maintained for coordination bonds with the paramagnetic metal ion, which limits the design options for new MRI contrast agents with customized T1/T2 performance.
Comments 2: Could the use of nanocarriers (such as liposomes) improve the water solubility and bioavailability of the composite, thereby enhancing in vivo imaging performance?
Response 2: Thank you for this comment. Previously published studies demonstrated that liposomes and other transport vehicles could significantly improve both aqueous solubility and bioavailability of insoluble in water drugs (Łopuszyńska, N.; Węglarz, W.P. Contrasting Properties of Polymeric Nanocarriers for MRI-Guided Drug Delivery. Nanomaterials 2023, 13, 2163. https://doi.org/10.3390/nano13152163; Crintea, A.; Dutu, A.G.; Sovrea, A.; Constantin, A.-M.; Samasca, G.; Masalar, A.L.; Ifju, B.; Linga, E.; Neamti, L.; Tranca, R.A.; et al. Nanocarriers for Drug Delivery: An Overview with Emphasis on Vitamin D and K Transportation. Nanomaterials 2022, 12, 1376. https://doi.org/10.3390/nano12081376). However, the optimization of the most suitable nanocarrier as a drug delivery system for the composites of salinomycin with 2−line ferrihydrite requires extensive research, and the data will be included in a separate manuscript.
Comments 3: What were the pharmacokinetics, plasma protein binding profile, and biodistribution of the composite in vivo? Could it achieve tumor-specific accumulation?
Response 3: Thank you for pointing this out. We would like to note that the aim of our current work was to assess the potential of our new composites as contrast agents and anticancer drugs in vitro (we revised the title of the manuscript and the text to clarify the aim of our research). Studies on animal models are out of the scope of our current project. We would like to note that the results presented in this manuscript demonstrate that the composites are suitable for further evaluation in vivo.
Comments 4: Under high-field (9.4T) and clinically common low-field (1.5-3T) MRI conditions, did r1 and r2 maintain similar trends?
Response 4: In MRI, both r1 and r2 relaxivities exhibit field-dependent behavior, but they do not always follow comparable trends across all magnetic field strengths. At lower magnetic field strengths, r1 relaxivity plateaus or even decreases, while r2 relaxivity normally increases with field strength (Shen, Y., Goerner, F. L., Snyder, C., Morelli, J. N., Hao, D., Hu, D., Li, X., & Runge, V. M. (2015). T1 relaxivities of gadolinium-based magnetic resonance contrast agents in human whole blood at 1.5, 3, and 7 T. Investigative radiology, 50(5), 330–338. https://doi.org/10.1097/RLI.0000000000000132; Rohrer, M., Bauer, H., Mintorovitch, J., Requardt, M., & Weinmann, H. J. (2005). Comparison of magnetic properties of MRI contrast media solutions at different magnetic field strengths. Investigative radiology, 40(11), 715–724. https://doi.org/10.1097/01.rli.0000184756.66360.d3). Rohrer et al. investigated how different magnetic field intensities affected the relaxivity of various MRI contrast media (CM). In their experiment, they compared the relaxivity of CM at 0.47 T, 1.5 T, 3 T, and 4.7 T in water and blood plasma, as well as in whole blood at 1.5 T. The findings showed that relaxivity is influenced by both field strength and solvent (water vs. plasma/blood) (Rohrer, M., Bauer, H., Mintorovitch, J., Requardt, M., & Weinmann, H. J. (2005). Comparison of magnetic properties of MRI contrast media solutions at different magnetic field strengths. Investigative radiology, 40(11), 715–724. https://doi.org/10.1097/01.rli.0000184756.66360.d3).
In our study, using a high-field magnetic strength of 9.4 T may be favorable for applying a reduced dose of contrast agent and achieving high-resolution images, but for the exact determination of r1 and r2 relaxivity values, several important factors should be considered, namely the choice of contrast agent, the kind of solvent, and the field strength. All these factors can be optimized for a specific imaging application.
Reviewer 2 Report
Comments and Suggestions for Authors
The article "First evaluation of two-line ferrihydrite composites with salinomycinic acid as theranostic probes for magnetic resonance imaging (MRI)" is a continuation of the work previously published by the authors in the article Ivanova, J.; et al. New Iron(III)-Containing Composite of Salinomycinic Acid with Antitumor Activity— Synthesis and Characterization. Inorganics 2024, 12, 206. However, the article should be clear to the reader without first reading the previous work. In particular, the authors do not cite either the structure of salinomycin or the proposed structures of the composites under study. Authors write «Herein, we investigated the potential of previously prepared two-line ferrihydrite composites of Sal as promising MRI contrast agents and anticancer drugs in one molecule» (lines 95-96). However, reference [18] to the previous work is missing. Therefore, it is not clear why composites 2-4 were used in the work. Why is composite 1 missing? The authors should briefly mention their previous findings and what additional information they wanted to obtain by conducting additional research.
The authors do not answer the question of how stable the composites they obtained will be under physiological conditions. In particular, in the presence of albumin, the main plasma protein capable of forming strong complexes with iron ions. The authors investigated relaxation in vitro. Since the authors consider their compounds as potential theranostics, it would also be logical to conduct studies of relaxation in blood plasma.
The authors note the low solubility of the resulting composites in water. What happens when organic solutions are diluted with water? Colloidal solutions formed? If so, what is the size of the colloidal particles? Without an answer to these questions, it is not clear what the authors worked with in determining cytotoxicity.
Since the answers to these questions require additional research, it may be necessary for the authors to change the title of the article and not present the resulting composites as potential theranostics for magnetic resonance imaging.
Author Response
Dear reviewer, we would like to thank you for your time to evaluate our manuscript. We appreciate your comments and provide the following responses below, along with the corresponding revisions in track changes in the resubmitted files.
Comments 1: However, the article should be clear to the reader without first reading the previous work. In particular, the authors do not cite either the structure of salinomycin or the proposed structures of the composites under study. Authors write «Herein, we investigated the potential of previously prepared two-line ferrihydrite composites of Sal as promising MRI contrast agents and anticancer drugs in one molecule» (lines 95-96). However, reference [18] to the previous work is missing. Therefore, it is not clear why composites 2-4 were used in the work. Why is composite 1 missing? The authors should briefly mention their previous findings and what additional information they wanted to obtain by conducting additional research.
Response 1: Thank you for pointing this out. We agree with these comments and revised the text with the appropriate explanations (please see the results section, first paragraph, text in track changes, lines 100-119). We have also added the structure of salinomycin in the Introduction section (lines 87-89). Since the structure of ferrihydrite is complicated and still debatable in literature (Michel, F.M.; Ehm, L.; Antao, S.M.; Lee, P.L.; Chupas, P.J.; Liu, G.; Strongin, D.R.; Schoonen, M.A.A.; Phillips, B.L.; Parise, J.B. The Structure of Ferrihydrite, a Nanocrystalline Material. Science 2007, 316, 1726–1729.), structures of composites of two−line ferrihydrite with organic compounds have never been reported (reference 18 in the manuscript).
Comments 2: The authors do not answer the question of how stable the composites they obtained will be under physiological conditions. In particular, in the presence of albumin, the main plasma protein capable of forming strong complexes with iron ions. The authors investigated relaxation in vitro. Since the authors consider their compounds as potential theranostics, it would also be logical to conduct studies of relaxation in blood plasma.
Response 2: According to your suggestion, we revised the manuscript, its title, and removed the term “theranostics”. Unfortunately, there is no suitable method to study the stability of the composites of 2−line ferrihydrite with salinomycin. We would like to point out that the formation and the presence of these particular composites could be proved only in the solid state by powder X−ray analysis. Molecular docking studies by Chilom et al. 2020, however, revealed low affinity of ferrihydrite nanoparticles to human serum albumin (Chilom CG, Bălan A, Sandu N, Bălăşoiu M, Stolyar S, Orelovich O. Exploring the Conformation and Thermal Stability of Human Serum Albumin Corona of Ferrihydrite Nanoparticles. Int J Mol Sci. 2020 Dec 20;21(24):9734. doi: 10.3390/ijms21249734.). We agree with your comment that extensive in vivo studies are needed to demonstrate the potent of the composites of salinomycin with 2−line ferrihydrite as theranostics. These studies require additional financial support, and the results will eventually be included in a separate manuscript.
Comments 3: The authors note the low solubility of the resulting composites in water. What happens when organic solutions are diluted with water? Colloidal solutions formed? If so, what is the size of the colloidal particles? Without an answer to these questions, it is not clear what the authors worked with in determining cytotoxicity.
Response 3: Thank you for pointing this out. Indeed, diluting the composites' solutions with MilliQ water prior to the MRI investigation, colloidal solutions were formed. We carried out additional DLS measurements to determine the hydrodynamic diameter and zeta potential of the three composites' solutions. Monodisperse particles with sizes between 338 and 420 nm and a low degree of agglomeration suggested by the zeta potential that lay between 19.2 and 22.0 mV, were detected. The additional DLS measurements were added with a method description and discussion of the obtained results in the corresponding parts of the revised manuscript. We would like to note that for MRI, the highest possible composite concentration was needed to prepare the dilution series and to assess the relaxivities in vitro. For the evaluation of cytotoxicity, the concentration lay in the submicromolar range so that no colloid formation occurred.
Moreover, to study the antitumor activity of the composites of salinomycin with 2−line ferrihydrite, we used standardized procedure, suitable for insoluble in water drugs, but soluble in DMSO (Petkov N, Pantcheva I, Ivanova A, Stoyanova R, Kukeva R, Alexandrova R, Abudalleh A, Dorkov P. Novel Cerium(IV) Coordination Compounds of Monensin and Salinomycin. Molecules. 2023 Jun 9;28(12):4676. doi: 10.3390/molecules28124676.). During the cell culture experiments, there were no observations pointing to instability of the composites or turbidity of the solution after adding cell culture medium.
Comments 4: Since the answers to these questions require additional research, it may be necessary for the authors to change the title of the article and not present the resulting composites as potential theranostics for magnetic resonance imaging.
Response 4: We agree with this comment and changed the title and revised the manuscript accordingly. The additional DLS measurements were added with a method description and discussion of the obtained results in the corresponding parts of the revised manuscript. The original DLS data are openly available in Mendeley Data at doi: 10.17632/p4wwk57mhf.2.
Round 2
Reviewer 1 Report
Comments and Suggestions for Authors
The authors have greatly improved the revised manuscript according to the comments point by point. I recommend its publication.